# *In Vitro* Activation of Human Adrenergic Receptors and Trace Amine-Associated Receptor 1 by Phenethylamine Analogues Present in Food Supplements

**DOI:** 10.3390/nu16111567

**Published:** 2024-05-22

**Authors:** Nicole E. T. Pinckaers, W. Matthijs Blankesteijn, Anastasiya Mircheva, Xiao Shi, Antoon Opperhuizen, Frederik-Jan van Schooten, Misha F. Vrolijk

**Affiliations:** 1Department of Pharmacology and Toxicology, Maastricht University, 6200 MD Maastricht, The Netherlands; 2Research Institute of Nutrition and Translational Research in Metabolism (NUTRIM), Maastricht University, 6200 MD Maastricht, The Netherlands; 3School for Cardiovascular Diseases (CARIM), Maastricht University, 6200 MD Maastricht, The Netherlands; 4Research Service, Veterans Affairs Portland Health Care System, Portland, OR 97239, USA; 5Department of Psychiatry, Oregon Health and Science University, Portland, OR 97239, USA; 6Office for Risk Assessment and Research, Netherlands Food and Consumer Product Safety Authority, 3540 AA Utrecht, The Netherlands

**Keywords:** food supplements, adrenergic receptors, trace amine-associated receptor 1, phenethylamine analogues, alkylamine analogues, adrenaline

## Abstract

Pre-workout supplements are popular among sport athletes and overweight individuals. Phenethylamines (PEAs) and alkylamines (AA) are widely present in these supplements. Although the health effects of these analogues are not well understood yet, they are hypothesised to be agonists of adrenergic (ADR) and trace amine-associated receptors (TAARs). Therefore, we aimed to pharmacologically characterise these compounds by investigating their activating properties of ADRs and TAAR1 *in vitro*. The potency and efficacy of the selected PEAs and AAs was studied by using cell lines overexpressing human ADRα_1A_/α_1B_/α_1D_/α_2a_/α_2B_/β_1_/β_2_ or TAAR1. Concentration–response relationships are expressed as percentages of the maximal signal obtained by the full ADR agonist adrenaline or the full TAAR1 agonist phenethylamine. Multiple PEAs activated ADRs (EC_50_ = 34 nM–690 µM; E_max_ = 8–105%). Almost all PEAs activated TAAR1 (EC_50_ = 1.8–92 µM; E_max_ = 40–104%). Our results reveal the pharmacological profile of PEAs and AAs that are often used in food supplements. Several PEAs have strong agonistic properties on multiple receptors and resemble potencies of the endogenous ligands, indicating that they might further stimulate the already activated sympathetic nervous system in exercising athletes via multiple mechanisms. The use of supplements containing one, or a combination of, PEA(s) may pose a health risk for their consumers.

## 1. Introduction

In recent years, the availability of food supplements with varying purposes on the market and their use are rising. In particular, pre-workout supplements and fat burners are becoming a more and more popular category of food supplements among sport athletes and overweight individuals [1] due to their advertised promises of improved sports performance and weight loss. Whereas a study on the use of such products in the Dutch society in 2012 showed that only 5.8% of regularly exercising participants were using sports enhancement supplements [2], a more recent study showed that 52.6% of the (amateur) sport athletes are using or have ever used such products [3]. This shows the substantial increase in the popularity of these nutritional supplements in the past few years.

Despite the large-scale use of these supplements, their adverse health effects are poorly investigated. Over the years, several cases of serious adverse health effects, like palpitations, myocardial infarction, cardiac arrest and brain haemorrhages, have been linked to the use of pre-workout and fat burner supplements [3,4,5,6,7,8,9]. Most of these adverse health effects are specifically associated with the cardiovascular system [10]. Since pharmacological and toxicological data about the ingredients of these supplements are often lacking, causally linking the adverse health effects to the use of the supplements and mechanistically explaining these effects remain challenging.

Most likely, exposure to the pharmacologically active substances (PASs) present in these supplements explains the evoked adverse health events. Various studies have identified the presence of known PASs and structurally related compounds in pre-workout and fat burner supplements, of which phenethylamine and alkylamine analogues constitute a large group [8,11,12,13,14,15,16,17,18,19,20,21,22,23]. Phenethylamines and alkylamines are two classes of molecules that have been shown to have psychoactive and stimulant effects [24,25]. These compounds are structurally similar to endogenous catecholamines, such as adrenaline, noradrenaline and dopamine (Figure 1a), and to exogenous stimulants like amphetamine. Catecholamines and amphetamines are both known to induce sympathetic activation via the stimulation of adrenergic receptors (ADRs) and the human trace amine-associated receptor 1 (TAAR1) located in the central nervous system (CNS) and the periphery [26,27,28]. Ultimately, these will both lead to increased activation of the cardiovascular system, which is physiologically shown by increased heart rate and blood pressure [29,30].

Remarkably, for most of the detected phenethylamines and alkylamines, no pharmacological profile is available. It is therefore not known if and to what extent these compounds interact with the TAAR1 and ADR and thus how they might affect the cardiovascular system. Despite this lack of knowledge, supplements containing these compounds are used on a large scale and often in high doses [3,11,19,31]. For example, intake levels of methylsynephrine were found to be two times higher than the maximum prescribed dosage of the former pharmaceutical Carnigen, which was formulated with methylsynephrine [19,32]. Besides the high dosages, this compound was present in a sport supplement that also contained other PASs (e.g., β-methylphenethylamine and dimethylphenethylamine), which might both lead to adverse health effects. It is therefore essential to gain more knowledge about the pharmacological profile of phenethylamine and alkylamine compounds that are often used as nutritional supplements. Therefore, this study aims to pharmacologically characterise a selection of these phenethylamine and alkylamine analogues *in vitro* by assessing their potency and efficacy to activate human ADRs, α_1_, α_2_, β_1_ and β_2,_ and TAAR1.

## 2. Materials and Methods

### 2.1. Selection of Phenethylamine and Alkylamine Analogues for Receptor Activation Assays

Search engines PubMed and Scopus were used to collect research papers on the detection of phenethylamine and alkylamine analogues in nutritional supplements. The key words/phrases used for the search were as follows: “pharmacologically active compounds supplements”, “bioactive compounds supplements”, “weight loss supplements”, “pre-workout supplements”, “fatburner”, “sport supplements”, “phenethylamine supplements”, “alkylamine supplements” and “amine supplements”. The most frequently reported analogues were selected to include in the in vitro pharmacological screening. The selected compounds are presented in Figure 1b,c [8,11,12,13,14,15,16,17,18,19,20,21,22,23].

### 2.2. Chemicals

Probenecid (purity ≥ 98%), adrenaline bitartrate, beta-methylphenethylamine (99%), phenethylamine (99.7%), tyramine hydrochloride (≥98%), halostachine (99%), hordenine (≥97.5%), *p*-octopamine (≥95%), dimethylbutylamine (98%), dimethylhexylamine (99%), dimethylaminoethanol (99.8%), puromycin dihydrochloride (10 mg/mL solution in mQ) and L-ascorbic acid (99%) were purchased from Sigma-Aldrich (St. Louis, MO, USA). *p*-synephrine hydrochloride (99.6%) and higenamine hydrochloride (98.98%) were purchased from MedChemExpress (Monmouth Junction, NJ, USA). Methyltyramine (97%) and 1,3-dimethylamylamine hydrochloride (95%) were purchased from Apollo Scientific (Stockport, UK). Sodium bicarbonate and isobutylmethylxanthine (IBMX) were purchased from Merck (Darmstadt, Germany). Fluo-8 AM (97%) was purchased from AAT Bioquest (Sunnyvale, CA, USA). Methylsynephrine hydrochloride (99.8%) was purchased from Mikromol (Luckenwalde, Germany), isopropyloctopamine acetate (99.9%) from Honeywell Fluka (Seetze, Germany), isopropyloctopamine hydrochloride (98%) from Toronto Research Chemicals (Toronto, ON, Canada) and N,N-dimethylphenethylamine hydrochloride (>99%) from GLPBio (Montclair, NJ, USA).

### 2.3. Adrenergic Receptor (ADR) Activation

Cells overexpressing human ADRs α_1A_, α_1B_, α_1D_, α_2A_, α_2B_, β_1_ and β_2_ (Eurofins Discovery Services, St. Charles, MO, USA, product numbers: HTS087RTA, HTS158RTA, HTS216RTA, HTS096RTA, HTS157LRTA, HTS104RTA and HTS073RTA, respectively) were used to study the potency (EC_50_) and efficacy (E_max_) of the selected compounds for these receptors by quantifying Ca^2+^ influx upon receptor activation. ADRα_1A_, α_1B_, α_1D_, α_2A_, β_1_ and β_2_ were overexpressed in rat chem-1 cells. ADRα_2B_ was overexpressed in HEK293 cells. The cells were seeded (10,000–50,000 cells/well) in a black 96-well plate (VWR International, Leuven, Belgium) and incubated at 37 °C and 5% CO_2_ for 24 h. After 24 h, the cells were washed with Hank’s balanced salt solution (HBSS) (Gibco, Paisley, UK) containing 0.35 g/L sodium bicarbonate and incubated with 5 µg/mL Fluo-8 AM in HBSS (pH 7.4) supplemented with 2.5 mM probenecid for 1 h. Baseline fluorescence (excitation: 490 nm; emission: 525 nm) was measured for 15 s with a read interval of 1.3 s, whereafter the selected test compounds were injected at a dispense rate of 78 µL/s to the cells using FlexStation 3 (Molecular Devices, Sunnyvale, CA, USA). After the addition of the test compounds, fluorescence was measured for another 165 s using the same settings. Peak tops of the obtained fluorescence response–time curves were corrected for baseline fluorescence and used to construct concentration–response relationships.

### 2.4. Trace Amine-Associated Receptor 1 (TAAR1) Activation

HEK293T cells stably expressing human TAAR1 (Appendix A), kindly donated by Aaron Janowsky, PhD from VA Portland Health Care System, were used to study the potency (EC_50_) and efficacy (E_max_) of the selected compounds for this receptor by quantifying intracellular cyclic adenosine monophosphate (cAMP) levels upon receptor activation. The cells were grown in Dulbecco’s Modified Eagle’s Medium with 4.5 g/L glucose and 584 mg/L L-glutamine (Sigma-Aldrich, St. Louis, MO, USA), supplemented with 10% foetal bovine serum (Gibco, Grand Island, NY, USA), 100 units/mL penicillin, 100 µg/mL streptomycin (Gibco, Grand Island, NY, USA), 26 mM sodium bicarbonate and 2 µg/mL puromycin, and were maintained in a humidified incubator with 5% CO_2_. The cells were seeded (90,000–250,000 cells/well) in a 48-well plate two days before the assay. Two hours before the assay, the cells were incubated with the culture medium containing 10% charcoal-stripped foetal bovine serum (Gibco, Grand Island, NY, USA). The assay was performed in Earle’s balanced salt solution (EBSS) (Gibco, Paisley, UK), containing 15 mM HEPES (Gibco, Paisley, UK), 1 mM IBMX and 0.02% ascorbic acid. First, the cells were pre-incubated for 20 min in the supplemented buffer. Subsequently, the cells were exposed to the test compounds for 1 h in an incubator at 37 °C with 5% CO_2_. After 1 h, the buffer was discarded and the cells were lysed with 3% trichloroacetic acid and incubated at room temperature on a plate shaker for 2 h. Cell lysates were further diluted and cAMP was measured using a cAMP ELISA kit (Cayman Chemical, Ann Arbor, MI, USA), according to the manufacturer’s protocol.

### 2.5. Data Analysis

Concentration–response relationships are shown as mean ± SD. For ADR activation, concentration–response relationships are presented as percentages of the maximal intracellular Ca^2+^ signal obtained by the full ADR agonist adrenaline and corrected for baseline measurements. Concentration–response relationships for TAAR1 activation are presented as percentages of the maximal intracellular cAMP level induced by full the TAAR1 agonist phenethylamine and corrected for baseline cAMP accumulation. Concentration–response curves were fitted to non-linear regression curves with a standard Hill slope of 1 (GraphPad Software, version 10.1.2, San Diego, CA, USA). Experiments were run in duplicate in 7–9 independent experiments for ADR and were run in 3–5 independent experiments for TAAR1. The EC_50_ and E_max_ values are presented as an average of the independent experiments, with a 95% confidence interval (95% CI).

## 3. Results

### 3.1. Adrenergic Receptor α_1A/B/D_ Activation

The concentration–response curves of the compounds that activated the three ADRα_1_ subtypes α_1A_, α_1B_ and α_1D_ are presented in Figure 2a,b,c, respectively. In addition, ADR-activating potency (EC_50_) and efficacy (E_max_) values of the selected compounds are presented in Table 1. Adrenaline activated ADRα_1A_, α_1B_ and α_1D_ with EC_50_ values of 6.5 nM, 6.1 nM and 1.7 nM, respectively. Four of the selected phenethylamines were found to activate ADRα_1A_. *p*-octopamine and *p*-synephrine activated ADRα_1A_ with EC_50_ values of 11 µM and 2.4 µM, respectively, and E_max_ values of 87% and 82%. Furthermore, halostachine and methylsynephrine activated ADRα_1A_ with EC_50_ values of 8.7 µM and 44 µM and E_max_ values of 59% and 16%, respectively. Similarly, five compounds activated ADRα_1B_ and ADRα_1D_. *p-O*ctopamine and *p*-synephrine activated ADRα_1B_ and ADRα_1D_ with EC_50_ values of 3.9 µM and 0.66 µM, respectivelyfor ADRα_1B_ (E_max_ = 94% and 91%); the EC_50_ values for ADRα_1D_ were 1.2 µM and 1.7 µM, respectively (E_max_ = 100% and 80%). Additionally, halostachine activated ADRα_1B_ and ADRα_1D_ with EC_50_ values of 1.1 µM and 2.1 µM, respectively, and acted as a partial agonist (E_max_ = 77% for ADRα_1B_ and E_max_ = 82% for ADRα_1D_). Both hordenine and dimethylphenethylamine acted as weak agonists for ADRα_1B_ and ADRα_1D,_ with EC_50_ values of 5.7 µM and 6.1 µM, respectively; for ADRα_1B_ (E_max_ = 37% and 19%), the values were 34 µM and 8.4 µM, respectively, and for ADRα_1D,_ E_max_ = 23% and 7.9%. The other phenethylamine analogues and all of the alkylamines did not activate these ADRs at concentrations up to 300 µM.

### 3.2. Adrenergic Receptor α_2A/B_ Activation

The concentration–response relationships of the compounds that activated ADRα_2A_ are presented in Figure 3. The concentration–response relationship of adrenaline at ADRα_2B_ is shown in Figure A2 in Appendix B. In addition, the activating potency (EC_50_) and efficacy (E_max_) values of the selected compounds are presented in Table 1. Adrenaline activated ADRα_2A_ with an EC_50_ value of 8.4 nM and ADRα_2B_ with an EC_50_ value of 23 nM. ADRα_2A_ was only activated by the compounds *p*-synephrine (EC_50_ = 100 µM and E_max_ = 89%) and hordenine (EC_50_ = 690 µM and E_max_ = 12%) (Figure 3). The other phenethylamine analogues and all of the alkylamines did not activate these ADRs at concentrations up to 300 µM.

### 3.3. Adrenergic Receptor β_1/2_ Activation

The concentration–response curves of the studied compounds that activated ADRβ_1_ and ADRβ_2_ are presented in Figure 4a,b. In addition, the activating potency (EC_50_) and efficacy (E_max_) values of the selected compounds are presented in Table 1. Adrenaline activated ADRβ_1_ with an EC_50_ of 9.9 nM and ADRβ_2_ with an EC_50_ of 67 nM. Five of the studied compounds were found to activate ADRβ_1_ and only one compound activated ADRβ_2_. Isopropyloctopamine and higenamine exhibited strong agonistic properties at ADRβ_1,_ with EC_50_ values of 117 nM and 34 nM, respectively (E_max_ = 105% for both). In comparison, *p*-octopamine, *p*-synephrine and methylsynephrine were less potent, with EC_50_ values of 5.5 µM, 28 µM and 25 µM, respectively (E_max_ = 88%, 64% and 75%). ADRβ_2_ was only partially activated by higenamine, with an EC_50_ value of 0.47 µM (E_max_ = 31%). The other phenethylamine analogues and all of the alkylamines did not activate these ADRs at concentrations up to 300 µM.

### 3.4. Trace Amine-Associated Receptor 1 (TAAR1) Activation

The concentration–response curves of the studied compounds that activated TAAR1 are presented in Figure 5. In addition, the activating potency (EC_50_) and efficacy (E_max_) values of the selected compounds are presented in Table 1. Phenethylamine activated TAAR1 with an EC_50_ value of 8.8 µM and an E_max_ value of 97%. Ten of the selected phenethylamine analogues were found to activate TAAR1. Higenamine, β-methylphenethylamine, tyramine and methyltyramine were found to be the most potent agonists, with EC_50_ values of 0.98 µM (E_max_ = 93%), 2.1 µM (E_max_ = 77%), 9.5 µM (E_max_ = 77%) and 23 µM (E_max_ = 83%), respectively. Isopropyloctopamine also had an EC_50_ value in this lower range (EC_50_ = 1.8 µM) but was least effective (E_max_ = 40%). In addition, dimethylphenethylamine, *p*-octopamine, hordenine, halostachine and *p*-synephrine activated TAAR1 with EC_50_ values of 21 µM (E_max_ = 64%), 46 µM (E_max_ = 85%), 47 µM (E_max_ = 82%), 74 µM (E_max_ = 104%) and 92 µM (E_max_ = 85%), respectively. Methylsynephrine did not activate TAAR1 at concentrations up to 300 µM. Alkylamine analogues 1,3-dimethylamylamine, dimethylbutylamine and dimethylhexylamine activated TAAR1 only at a concentration of 300 µM, with an effect of 52%, 20% and 36%, respectively, of the maximal effect obtained by phenethylamine (see Figure A3 in Appendix C). Dimethylaminoethanol activated the receptor with an effect of approximately 5% of the maximal effect obtained by phenethylamine.

**Table 1 nutrients-16-01567-t001:** Human adrenergic receptor (ADR)- and trace amine-associated receptor 1 (TAAR1)-activating potency (EC_50_) and efficacy (E_max_) of adrenaline and the selected phenethylamine and alkylamine analogues. E_max_ is the maximal signal obtained by the full ADR agonist adrenaline or the full TAAR1 agonist phenethylamine. Compounds with a dash (–) showed no effects at concentrations up to 300 µM (n = 3). Compounds with an * showed only an effect at 300 µM and their effects at 300 µM are presented in the E_max_ column as % of the maximal effect obtained by phenethylamine. Values are shown as an average (95% confidence interval) of 7–9 independent experiments with 2 technical replicates for ADR and 3–5 independent experiments with 2 technical replicates for TAAR1.

	ADRα_1A_		ADRα_1B_		ADRα_1D_		ADRα_2A_		ADRβ_1_		ADRβ_2_		TAAR1	
	EC_50_ (M)	E_max_ (%)	EC_50_ (M)	E_max_ (%)	EC_50_ (M)	E_max_ (%)	EC_50_ (M)	E_max_ (%)	EC_50_ (M)	E_max_ (%)	EC_50_ (M)	E_max_ (%)	EC_50_ (M)	E_max_ (%)
Adrenaline	6.5 × 10^−9^ (6.0–7.0 × 10^−9^)	98 (98–98)	6.1 × 10^−10^ (5.6–6.7 × 10^−10^)	99 (98–99)	1.7 × 10^−9^ (1.5–1.9 × 10^−9^)	99 (97–100)	8.4 × 10^−9^ (7.4–9.4 × 10^−9^)	105 (104–106)	9.9 × 10^−9^ (3.6–5.9 × 10^−9^)	99 (97–100)	6.7 × 10^−8^ (5.4–7.9 × 10^−8^)	104 (103–105)		
**Phenethylamines**														
Phenethylamine	-	-	-	-	-	-	-	-	-	-	-	-	8.8 × 10^−6^ (6.8–11 × 10^−6^)	97 (96–99)
β-Methylphenethyl-amine	-	-	-	-	-	-	-	-	-	-	-	-	2.1 × 10^−6^ (1.9–2.3 × 10^−6^)	77 (74–80)
Tyramine	-	-	-	-	-	-	-	-	-	-	-	-	9.5 × 10^−6^ (5.3–14 × 10^−6^)	77 (75–80)
Methyltyramine	-	-	-	-		-	-	-	-	-	-	-	2.3 × 10^−5^ (1.8–2.9 × 10^−5^)	83 (78–87)
*p*-Synephrine	2.4 × 10^−6^ (2.1–2.7 × 10^−6^)	82 (79–84)	6.6 × 10^−7^ (5.8–7.4 × 10^−7^)	91 (89–92)	1.7 × 10^−6^ (1.4–1.9 × 10^−6^)	80 (78–81)	1.0 × 10^−4^ (0.87–1.2 × 10^−4^)	89 (83–94)	2.8 × 10^−5^ (2.6–3.0 × 10^−6^)	64 (59–69)	-	-	9.2 × 10^−5^ (7.8–11 × 10^−5^)	85 (82–87)
Methylsynephrine	4.4 × 10^−5^ (3.5–5.3 × 10^−5^)	16 (14–18)	-	-	-	-	-	-	2.5 × 10^−5^ (2.3–2.8 × 10^−5^)	75 (70–80)	-	-	-	-
Higenamine	-	-	-	-	-	-	-	-	3.4 × 10^−8^ (2.8–4.0 × 10^−8^)	105 (102–108)	4.7 × 10^−7^ (4.2–5.3 × 10^−7^)	31 (29–33)	9.8 × 10^−7^ (8.4–11 × 10^−7^)	93 (80–106)
Halostachine	8.7 × 10^−6^ (7.1–10 × 10^−6^)	59 (57–60)	1.1 × 10^−6^ (0.98–1.2 × 10^−6^)	77 (75–80)	2.1 × 10^−6^ (1.9–2.3 × 10^−6^)	82 (79–85)	-	-	-	-	-	-	7.4 × 10^−5^ (6.4–8.4 × 10^−5^)	100 (94–114)
Hordenine	-	-	5.7 × 10^−6^ (5.3–6.0 × 10^−6^)	37 (34–39)	3.4 × 10^−5^ (2.2–4.5 × 10^−5^)	23 (20–25)	6.9 × 10^−4^ (3.7–10 × 10^−4^)	28 (22–33)	-	-	-	-	4.7 × 10^−5^ (3.5–5.8 × 10^−5^)	82 (78–86)
*p-O*ctopamine	1.1 × 10^−5^ (0.93–1.2 × 10^−5^)	87 (86–89)	3.9 × 10^−6^ (3.5–4.3 × 10^−6^)	94 (92–96)	1.2 × 10^−6^ (1.0–1.3 × 10^−6^)	100 (99–102)	-	-	5.5 × 10^−6^ (5.0–6.0 × 10^−6^)	88 (87–90)	-	-	4.6 × 10^−5^ (3.8–5.3 × 10^−5^)	85 (77–92)
Isopropyloctopamine	-	-	-	-	-	-	-	-	1.2 × 10^−7^ (0.99–1.4 × 10^−7^)	105 (100–109)	-	-	1.8 × 10^−6^ (0.95–2.6 × 10^−6^)	40 (35–45)
Dimethylphenethylamine	-	-	6.1 × 10^−6^ (5.4–6.9 × 10^−6^)	19 (17–21)	8.4 × 10^−6^ (7.3–9.6 × 10^−6^)	7.9 (7.3–8.5)	-	-	-	-	-	-	2.1 × 10^−5^ (1.2–2.9 × 10^−5^)	64 (59–70)
**Alkylamines**														
1,3-Dimethylamylamine	-	-	-	-	-	-	-	-	-	-	-	-	*	55 (50–61)
Dimethylbutylamine	-	-	-	-	-	-	-	-	-	-	-	-	*	20 (17–23)
Dimethylhexylamine	-	-	-	-	-	-	-	-	-	-	-	-	*	5.3 (3.4–7.2)
Dimethylaminoethanol	-	-	-	-	-	-	-	-	-	-	-	-	*	36 (34–38)

## 4. Discussion

In this study, the pharmacological profile of a group phenethylamine and alkylamine analogues, which are frequently detected in sport and weight loss supplements [8,11,12,13,14,15,16,17,18,19,20,21,22,23], was determined by assessing the receptor-activating potencies and efficacies at human ADRα_1A_, α_1B_, α_1D_, α_2a_, α_2B_, β_1_ and β_2_ and TAAR1. These receptors play a key role in the homeostatic regulation of the (cardio)vascular system, mainly by means of (in)direct stimulation of the sympathetic nervous system (SNS), which regulates heart rate and blood pressure [28,30]. During rest, the SNS and the parasympathetic nervous system (PNS) are tightly kept in balance by performing the opposite physiological functions [33]. In a state of exercise, this balance shifts toward the SNS, which becomes more activated in order to increase the oxygen supply in the exercising muscles. To achieve this increased oxygen delivery, the respiratory and cardiovascular system are stimulated directly by ADRs and indirectly by TAAR1. Our results show that multiple phenethylamine analogues are pharmacologically active at these receptors, which suggests that they can activate the SNS in addition to exercise.

ADRs selectively bind the catecholamines noradrenaline and adrenaline after their release from sympathetic nerve endings or the adrenal gland. This interaction leads to an intracellular signalling cascade that ultimately results in a physiological effect, such as vascular smooth muscle contraction and increased heart rate, both being critically involved in cardiac function and blood pressure homeostasis. ADRs can generally be classified into α_1_, α_2_, β_1_ and β_2_ receptors, each having different roles in the regulation of cardiovascular homeostasis [34].

The α ADRs modulate sympathetic activity in the central and peripheral nervous system and thereby control cardiovascular function. ADRα_1_ is expressed in vascular smooth muscle cells and plays a major role in blood pressure regulation by mediating vascular smooth muscle contraction [35]. The analogues that activated ADRα_1_ in the present study were mainly partial agonists and were more than hundred times less potent than the endogenous agonist adrenaline. *p*-Synephrine and *p*-octopamine were the most effective and potent phenethylamines on the ADRα_1_ subtypes in this study, with EC_50_ values more than 1000 times higher than reported plasma concentrations in humans after single oral doses of 20–50 mg *p*-synephrine [36,37]. Haller et al. (2005) showed an increase in both systolic and diastolic blood pressure in humans after consumption of a weight loss supplement containing a mixture of PASs, including *p*-synephrine [36]. Remarkably, increasing the dose of *p*-synephrine in this mixture did not alter the blood pressure, which may imply that *p*-synephrine alone was not responsible for the increase in blood pressure. Another explanation could be that *p*-synephrine is not a selective ADR α agonist and that its ADRα_1_ effects may be counteracted by the activation of central ADRα_2A_. However, it is difficult to causally link any of the observed effects to one of the ingredients, as the nutritional supplement also contained other PASs that were known for their cardiovascular side effects [38]. This emphasises the need for research approaches that study single-ingredient effects before studying mixture pharmacology. So far, no studies have been reported on the effects of *p*-octopamine on blood pressure in humans [39,40]. Pharmacodynamic and pharmacokinetic research of most of the studied phenethylamines is scarce. A few pharmacokinetic studies on the partial ADRα_1_ agonist hordenine after oral consumption in humans, with limited subject numbers (n = 3), are published. Human plasma levels of 2.7 nM and 15 nM after a 100 mg and 5.5 mg oral dose, respectively, were previously reported [41,42]. These oral doses did not result in plasma concentrations in the same range as the EC_50_ values in our study, which are three orders of magnitude higher. However, Biesterbos et al. (2019) reported a maximal daily dose of 669 mg hordenine in nutritional supplements, which indicates that the hordenine intake via nutritional supplements could be higher [11]. To date, no studies on the effects of hordenine on the cardiovascular system in humans have been published. In horses, hordenine was found to activate the sympathetic nervous system (SNS), as it increased heart rate and respiratory rate [43]. Hence, our study suggests that hordenine could have effects on blood pressure in humans, shown by its activating properties on ADRα_1B_ and ADRα_1D_.

Our study also pharmacologically characterised the selected phenethylamines and alkylamines on two ADRα_2_ subtypes: ADRα_2A_ and ADRα_2B_. The α_2_ ADR is involved in the regulation of peripheral vascular resistance. None of the studied phenethylamines and alkylamines activated ADRα_2B_. Only *p*-synephrine and hordenine were able to partially activate ADRα_2A,_ with potencies that were >12000 times lower compared to the endogenous agonist adrenaline. Hordenine showed an efficacy smaller than 30% compared to adrenaline and therefore is unlikely to exhibit major effects on blood pressure control via ADR α_2A_. *p-s*ynephrine had an efficacy of 89% and is therefore more likely to cause physiological effects. ADRα_2_ activation is shown to result in divergent physiological effects, depending on the site of action. The activation of ADRα_2A_ in the CNS decreases blood pressure by inhibiting synaptic noradrenaline release. On the other hand, ADRα_2A_ in vascular smooth muscle cells causes smooth muscle cell contraction, leading to vasoconstriction and, in this way, increases blood pressure [44,45].

Besides ADRα_1_ and ADRα_2_, the activating effects of the phenethylamine analogues were also screened for β_1_ ADRs, which play a major role in regulating cardiac contractility and heart rate. Similarly to α_1_ and α_2_, *p*-synephrine also acted as an ADRβ_1_ agonist. The addition of a methyl group to the carbon chain of *p*-synephrine did not alter its efficacy and potency on ADRβ_1_, as methylsynephrine showed similar results. Both compounds were approximately 2500 times less potent than the endogenous agonist adrenaline. *p*-Octopamine was slightly more effective and >500 times less potent than adrenaline on this receptor subtype. Both isopropyloctopamine and higenamine acted as full agonists and were found to have potency values in the same order of magnitude as adrenaline (approximately 10 and 3 times less potent than adrenaline). As ADRβ_1_ is predominantly located on cardiomyocytes in the heart, these findings suggest that these compounds may increase cardiac contraction force and rate [46]. This is also shown by Feng et al. (2012) who established a significant increased heart rate from 68 bpm at baseline to 126 bpm during a maximal plasma concentration of 8.4 nM higenamine in humans [47]. Case reports have revealed sympathetic overstimulation (leading to adverse effects such as palpitations, chest pain, tachycardia and even cardiac arrest) after consumption of multi-ingredient weight loss supplements containing isopropyloctopamine [8,17]. An increased heart rate was also observed after intravenous administration of isopropyloctopamine in subjects pre-treated with the non-selective ADRβ antagonist propranolol [48], which suggests that isopropyloctopamine competes with this ADRβ antagonist for receptor binding. These reports clearly highlight the cardiovascular risks of using sympathomimetic compounds like isopropyloctopamine in relatively high doses.

The β_2_ ADR plays an important in role bronchodilation and the regulation of vascular homeostasis by mediating vascular smooth muscle relaxation and it contributes to an increased heart contraction force and rate after activation in the heart [49,50]. Compared to ADRβ_1,_ it less intensively contributes to this latter effect as the ratio of ADRβ_1_ and ADRβ_2_ is 77% to 23%, respectively [51]. Higenamine was the only analogue found to act as a partial ADRβ2 agonist in the present study and was found to be seven times less potent than the full agonist adrenaline. Feng et al. (2012) observed that higenamine administration led to a slight decrease in diastolic blood pressure in humans, which could be explained by ADRβ_2_-mediated vasodilatation [47].

TAAR1 is expressed in both brain and peripheral tissues (e.g., blood vessels and the heart), where it is suggested to play a role in both direct and indirect activation of the SNS [27,28,52,53]. In the brain, TAAR1 is involved in stimulating synaptic neurotransmitter release and in synaptic neurotransmitter uptake inhibition [54], which both indirectly activate the SNS. Peripheral TAAR1 is found to indirectly activate the SNS by stimulating noradrenaline release in nerve endings and by a direct effect on vasoconstriction [55]. Broadley and Mehta (2023) recently proposed that phenethylamine-induced vasoconstriction in human, pig and rat blood vessels was a direct effect of TAAR1, as an ADR α1 antagonist did not inhibit vasoconstriction [28,56,57,58]. In the present study, ten of the studied phenethylamine analogues acted as TAAR1 agonists, with their efficacy ranging from 40 to 100%. Strikingly, β-methylphenethylamine, isopropyloctopamine and higenamine were more potent than the endogenous agonist phenethylamine. The other studied phenethylamine analogues were less potent than phenethylamine, but the EC50 values were still within a 10-fold change. Phenethylamine, tyramine, *p*-octopamine and *p*-synephrine are endogenous agonists of TAAR1 that are present in the CNS at concentrations more than hundred times lower than their traditional analogues (e.g., dopamine, serotonin and noradrenaline) and are therefore called “trace amines” [59]. Although the exact (patho)physiological role of TAAR1 is not known yet, the use of TAAR1 agonists is linked to severe cardiovascular health effects. Supplemental use of β-methylphenethylamine is previously linked to haemorrhagic strokes in exercising individuals [4,5]. Additionally, β-methylphenethylamine also elevated blood pressure in rats through indirect activation of the SNS with a similar efficacy to its isomer amphetamine, which is a well-known TAAR1-agonist [60,61]. These amphetamine-like effects caused by higher concentrations of trace amines (reached, for instance, by means of supplemental intake) are different from the effects caused by the endogenous trace amines at concentrations a hundred to a thousand times lower. At high concentrations, the activation of TAAR1 in the CNS will affect neurotransmission by increasing neurotransmitter release and inhibiting the reuptake of dopamine, noradrenaline and serotonin. This will consequently lead to overactivation of the SNS and the consequential overstimulation of the cardiovascular system [54,59,62]. These data show that the use of phenethylamines in high doses via supplements may stimulate the cardiovascular system and thereby pose a significant risk to human health.

In order to exert effects via TAAR1 localised in the brain, agonists need to cross the blood–brain barrier and pass through the cell membrane, as TAAR1—unlike other G protein-coupled receptors—is localised in the cytoplasm [61]. Tyramine and phenethylamines have previously been shown to pass through lipid bilayers by means of passive diffusion [63]. Additionally, many trace amines have a high affinity for Organic Cation Transporter 2 (OCT2) and dopamine, noradrenaline and serotonin transporters [27,64] and are therefore expected to reach intracellular TAAR1.

The alkylamine analogue 1,3-dimethylamylamine exclusively and partially activated TAAR1, but only at a concentration of 300 µM, which is too high to be physiologically relevant. The other alkylamine analogues also activated TAAR1 only at 300 µM but were even less effective. Neither of the alkylamine analogues activated ADRs at concentrations up to 300 µM. However, multiple case studies describe a link between adverse cardiovascular events and the consumption of pre-workout supplements containing these substances [65,66,67]. These studies suggest that also alkylamines might have sympathomimetic effects after consumption. Our current study shows that these effects cannot be mechanistically explained by their interaction with ADRs and probably not either by TAAR1. Recent studies have indeed proposed different pharmacological mechanisms for these types of substances, namely, their interaction with noradrenaline and dopamine transporters in the brain [25,68].

The links between cardiovascular toxicity and the use of weight loss and pre-workout supplements are in line with the historical evolution of the development of obesity medication [69]. Sympathomimetics are particularly of interest for weight loss medication as part of obesity treatment [70]. The application of sympathomimetics in weight loss management has extensively been studied in the past. Nevertheless, only a few drugs are currently approved (e.g., phentermine/topiramate, liraglutide, semaglutide, naltrexone/bupropion and orlistat), which are all indirectly acting sympathomimetics [71]. Cardiovascular side effects have been the main reason for the withdrawal of most of these drugs [69,71,72]. In particular, sympathomimetics (e.g., sibutramine, phentermine, methamphetamine and fenfluramine) that inhibit neurotransmitter reuptake or stimulate neurotransmitter release have failed to fulfil cardiovascular safety standards [69,72]. Despite the fact that they have failed to be successful for safe weight loss, a lot of these withdrawn sympathomimetics are nowadays drugs of abuse (e.g., in weight loss and pre-workout supplements) [73]. In this study, we showed that phenethylamine analogues are another potential group of compounds of abuse, as they could potentially act as sympathomimetics by activating ADRs or TAAR1.

## 5. Conclusions

Taken together, our results reveal the pharmacological activity of a selection of phenethylamine analogues regarding ADRs (α_1_, α_2_, β_1_ and β_2_) and TAAR1. Supplements containing these compounds are highly popular among (amateur and professional sports) individuals to improve their performance and/or to lose weight. Many of the phenethylamine analogues were shown to be (potent) activators of ADRs and TAAR1. Remarkably, some of the compounds (isopropyloctopamine and higenamine) have a potency comparable to the endogenous agonists for the receptor. Moreover, several phenethylamines have agonistic activity on multiple of these receptors (Table 1), indicating that they might further stimulate the already activated SNS in exercising individuals through multiple ways. For instance, higenamine is a potent agonist for both TAAR1 and ADR β_1_ and *p*-synephrine is a potent agonist for α_1_, α_2_, β_1_ and TAAR1, meaning that these compounds could affect the cardiovascular system directly via the ADRs and TAAR1 in the heart and vasculature and indirectly via TAAR1 in the brain. These individual effects may both contribute to an amplified combined effect on the cardiovascular system and may thereby adversely affect cardiovascular parameters like blood pressure and heart rate. Besides this, users may not always be aware of the presence of phenethylamine or its analogues, since for many of these supplements, the names of the plant-derived extracts are listed on the label, like *Citrus aurantium* (bitter orange), *Acacia rigidula*, *Hordeum vulgare* (barley) or *Ariocarpus retusus* (cactus), which are natural sources of phenethylamine and many of its studied analogues [22,74,75,76,77]. Consequently, the use of supplements containing one, or a combination of, phenethylamine(s) may pose a serious health risk to their consumers, especially when used by obese individuals with an already increased risk of cardiovascular diseases [78]. The collection of case reports on people using these supplements containing PASs indeed confirms this health risk, which is in line with the proven history of the development of obesity medication and its corresponding cardiotoxicity. However, since pharmacodynamic and pharmacokinetic information about the ingredients of these supplements is often lacking, more research needs to focus on the physiological effects of these compounds in humans.

## Figures and Tables

**Figure 1 nutrients-16-01567-f001:**
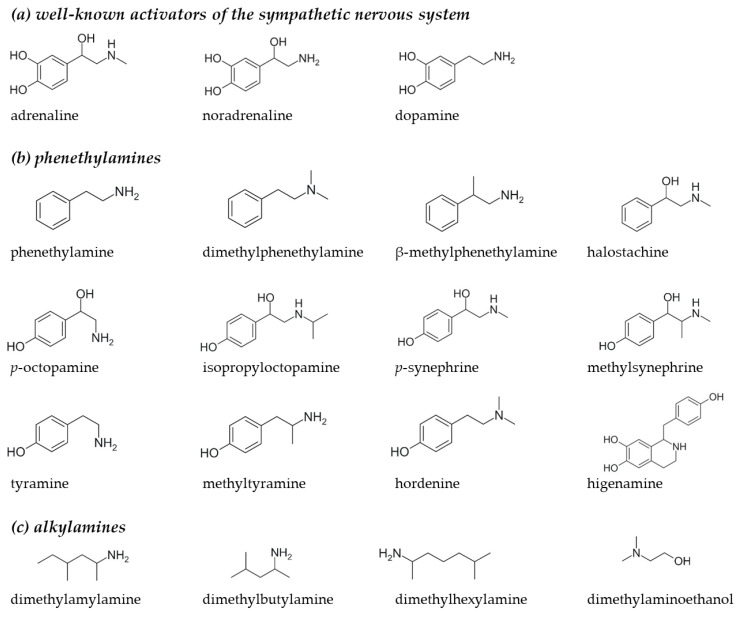
Molecular structures of (**a**) well-known activators of sympathetic nervous system: adrenaline, noradrenaline and dopamine, (**b**) phenethylamines and (**c**) alkylamines present in nutritional supplements.

**Figure 2 nutrients-16-01567-f002:**
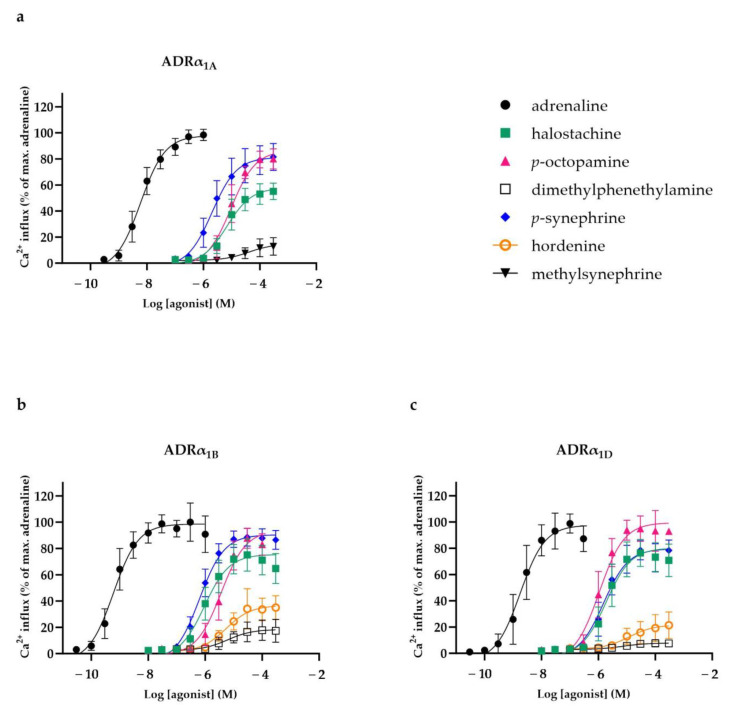
Concentration–response relationships (mean ± SD) of compounds that activated human adrenergic receptor (ADR) α_1A_ (**a**), α_1B_ (**b**) and α_1D_ (**c**) in chem-1 cells overexpressing these receptors. Data are presented as percentages of maximal fluorescent Ca^2+^ response obtained by adrenaline (n = 7–9).

**Figure 3 nutrients-16-01567-f003:**
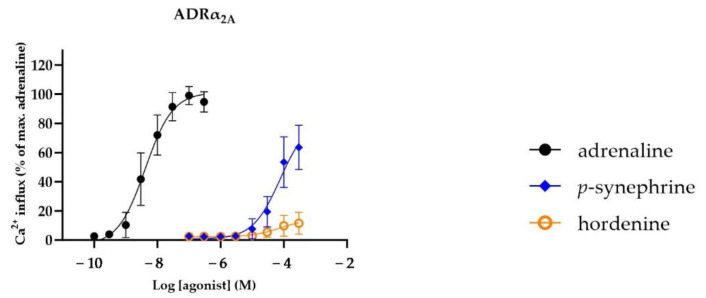
Concentration–response relationships (mean ± SD) of compounds that activated human adrenergic receptor (ADR) α_2A_ in chem-1 cells. Data are presented as percentages of maximal fluorescent Ca^2+^ response obtained by adrenaline (n = 8).

**Figure 4 nutrients-16-01567-f004:**
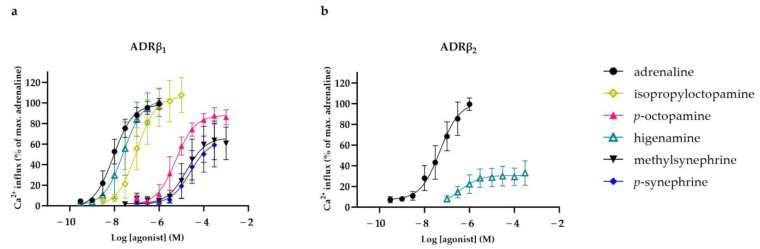
Concentration–response relationships (mean ± SD) of compounds that activated human adrenergic receptor (ADR) β_1_ (**a**) and β_2_ (**b**) in chem-1 cells overexpressing these receptors. Data are presented as percentages of maximal fluorescent Ca^2+^ response obtained by adrenaline (n = 8).

**Figure 5 nutrients-16-01567-f005:**
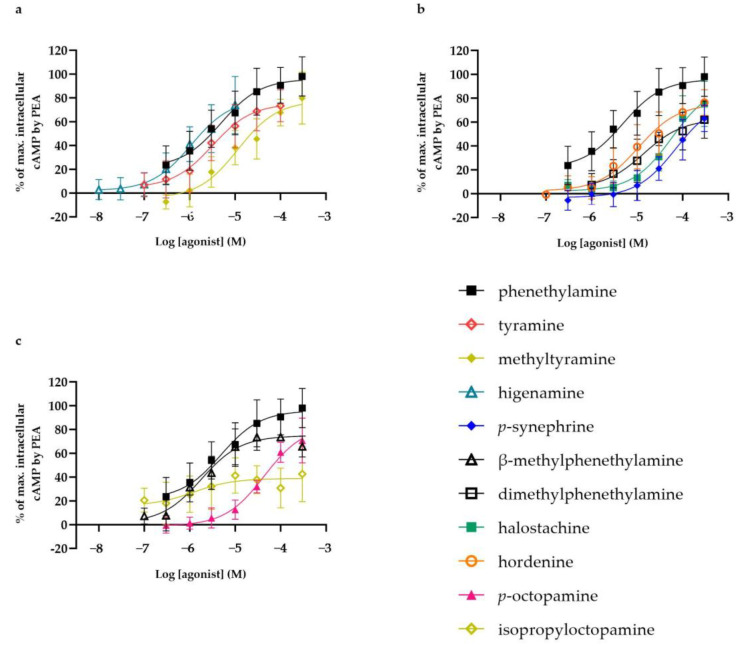
Concentration–response relationships (mean ± SD) of (**a**) phenethylamine, tyramine, methyltyramine and higenamine, (**b**) phenethylamine, *p*-synephrine, hordenine, dimethylphenethylamine and halostachine, and (**c**) phenethylamine, β-methylphenethylamine, *p*-octopamine and isopropyloctopamine, which activated human trace amine-associated receptor 1 (TAAR1) in HEK293T cells stably transfected with this receptor. Data are presented as percentages of maximal intracellular cAMP level obtained by phenethylamine (n = 3–5).

## Data Availability

The original contributions presented in this study are included in the article; further inquiries can be directed to the corresponding author.

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
