# Peer review of "In Vitro Activation of Human Adrenergic Receptors and Trace Amine-Associated Receptor 1 by Phenethylamine Analogues Present in Food Supplements"

_nutrients, 2024, doi:10.3390/nu16111567_

Round 1

Reviewer 1 Report

Comments and Suggestions for Authors

The idea, design, and execution of experiments investigating the agonistic effect of the tested compounds on adrenergic (ADR) and trace amine associated receptors (TAARs) seem correct. The rationale for taking up this research topic is also for people's cardiac health. However, in order to improve the quality of the manuscript, it would be worthwhile to make a few additions:

1.       The main doubt of these studies is the justification of the used concentrations of the tested compounds. In the Introduction to the manuscript, there is a sentence "Despite this lack of knowledge, supplements containing these compounds are used on a large scale and often in high doses [3,11,31], which might lead to adverse health effects.'. This information is too vague and does not convince the reader of the necessity of undertaking the study. Please provide more details.

2.       The concentrations of the tested compounds were significantly higher than those observed in plasma in humans after their single ingestion. Therefore, there is a lack of hard data on the pharmacokinetics of these compounds. In modelling studies, higher concentrations of compounds are used to show their mechanism of action. However, the question remains: is it possible to obtain concentrations similar to those used in the present studies after long-term supplementation with these compounds? Please refer to this point in detail to justify the reasonableness of the research undertaken.

3.       Please include in supplementary data a result confirming the overexpressing of human ADRs and TAAR1 in cells.

Reviewer 2 Report

Comments and Suggestions for Authors

Review of nutrients-2985653-peer review-v1

In the article, Phenethylamines (PEAs) and Alkylamines (AA) were evaluated in terms of their use in pre-workout supplements for sport athletes and overweight individuals.

they are hypothesized to be agonists of adrenergic (ADR) and trace amine associated receptors (TAARs). Potency and efficacy of the selected PEAs and AAs was studied by using cell lines overexpressing human ADRα1A/α1B/α1D/α2a/α2B/β1/β2 or TAAR1. Concentration-response relationships are expressed as percentages of the maximal signal obtained by the full ADR agonist adrenaline or the full TAAR1 agonist phenethylamine. Multiple PEAs activated ADRs (EC50=34 nM 26 – 690 μM; Emax=8 – 105%). Almost all PEAs activated TAAR1 (EC50=1.8 – 92 μM; Emax=40 – 104%). Based on studied PEAs and AA a health risk for consumers is discussed.

To the article, I have next comments and recommendations:

  • The significance of this study lies in the fact that it brings new, previously unknown data about PEAs and AA with their Emax and EC50 strong agonistic properties and overdose possibilities with negative effects such as increased heart rate, palpitations, chest pain, tachycardia and even cardiac arrest or high blood pressure. Regarding the data of this study high activating potency EC50 and efficacy show p-synephrine, hordenine and dimethylphenethylamine.

  • Perhaps it would be useful for readers to add sources to the article, e.g., for hordenine Hordeum vulgare, seaweed, Aconitum tanguticum (Maxim.) Stapf, Senecio scandens, Coryphantha ramillosa and Citrus aurantium, synthesis; for p-synephrine; for p-octoamine bitter orange (Citrus aurantium); for halostachine Halostachys belangeriana; for N,N-Dimethylphenethylamine (N,N-DMPEA), the orchid (Eria jarensis)  etc.

  • Throughout the article should be corrected: p- should be in Italics, p-synephrine, p-octoamine, also in tables and figures. When it is at the beginning of a sentence or in a table, should be p-Synephrine, p-Octoamine; similarly, β-Methylphenethyl-amine. L. 467: Ca2+.

  • References: should be unified titles of journals: full titles or their abbreviations according to journal´s recommendation, not their mixture. It is also necessary to unify before the last author: either…., author or…,  author, again not their mixture. Also, Latin names of plants should be in Italics, e.g., Citrus aurantium.
